# Tracking of Multiple Static and Dynamic Targets for 4D Automotive Millimeter-Wave Radar Point Cloud in Urban Environments

**Bin Tan** [1]  , **Zhixiong Ma** [1,*]  , **Xichan Zhu** [1], **Sen Li** [1]  , **Lianqing Zheng** [1], **Libo Huang** [2] **and Jie Bai** [2]

[1]   The School of Automotive Studies, Tongji University, Shanghai 201804, China
[2]   The School of Information and Electricity, Zhejiang University City College, Hangzhou 310015, China
*   Correspondence: mzx1978@tongji.edu.cn

**Abstract:** This paper presents a target tracking algorithm based on 4D millimeter-wave radar point cloud information for autonomous driving applications, which addresses the limitations of traditional 2 + 1D radar systems by using higher resolution target point cloud information that enables more accurate motion state estimation and target contour information. The proposed algorithm includes several steps, starting with the estimation of the ego vehicle's velocity information using the radial velocity information of the millimeter-wave radar point cloud. Different clustering suggestions are then obtained using a density-based clustering method, and correlation regions of the targets are obtained based on these clustering suggestions. The binary Bayesian filtering method is then used to determine whether the targets are dynamic or static targets based on their distribution characteristics. For dynamic targets, Kalman filtering is used to estimate and update the state of the target using trajectory and velocity information, while for static targets, the rolling ball method is used to estimate and update the shape contour boundary of the target. Unassociated measurements are estimated for the contour and initialized for the trajectory, and unassociated trajectory targets are selectively retained and deleted. The effectiveness of the proposed method is verified using real data. Overall, the proposed target tracking algorithm based on 4D millimeter-wave radar point cloud information has the potential to improve the accuracy and reliability of target tracking in autonomous driving applications, providing more comprehensive motion state and target contour information for better decision making.

**Keywords:** target tracking; 4D millimeter-wave radar; motion state estimation; autonomous driving

## 1. Introduction

For autonomous driving systems, accurately sensing the surrounding environment is crucial. Among the various vehicle sensing sensors, millimeter-wave radar is capable of obtaining position and speed information of targets, and can operate in complex weather conditions such as rain, fog, and bright sunlight exposure [1].

Conventional 2 + 1D (x, y, v) millimeter-wave radar is effective in measuring the radial distance, radial velocity, and horizontal angular information of a target. However, when compared to cameras and LIDAR, which are the other major sensors used in autonomous driving, traditional millimeter-wave radar has a lower angular resolution and cannot provide height angle information of the target. In autonomous driving scenarios, where vehicle or pedestrian targets are common, the small number of points and low angular resolution of individual targets in the scene can result in large errors in size and location estimation. To address this issue, high-resolution 4D (x, y, z, v) millimeter-wave radar has been developed, which can provide height angle information of targets with higher angular resolution. This enables more accurate edge information of targets and more precise estimation of a target's size and position.

Radar target tracking plays a critical role in millimeter-wave radar sensing. By providing a continuous position and velocity profile of a target, radar target tracking offers higher accuracy and reliability compared to a single measurement from the radar. Furthermore, it can effectively eliminate false detections.

Most conventional millimeter-wave radar tracking methods focus on point targets, which provide target ID, position, and velocity information. However, 4D millimeter-wave radar can measure multiple scattering centers per target, making direct application of point cloud tracking methods unsuitable. In addition, contour information, such as target size and orientation, is critical in autonomous driving environments. Therefore, accurate estimation of target ID, position, size, direction, and velocity information is necessary for 4D millimeter-wave radar target tracking. Despite considerable research on point target tracking using millimeter-wave radar, there is limited research on 4D millimeter-wave radar-based target tracking methods. Dynamic target tracking using 4D millimeter-wave radar presents several challenges, including variation in target size and multiple individual target measurement points. Furthermore, 4D millimeter-wave radar can measure static targets in the scene, while conventional millimeter-wave radar usually filters out static targets due to the absence of altitude angle information, which can result in false positives. Therefore, 4D millimeter-wave radar target tracking can also estimate the contour shape information of static targets. This paper focuses on developing tracking methods for multiple dynamic and static targets throughout a scene using 4D millimeter-wave radar.

The most commonly used multi-target tracking methods for millimeter-wave radar based on point targets include nearest neighbor data association (NN) [2,3], global nearest neighbor association (GNN) [4,5], multiple hypothesis tracking (MHT) [6,7], joint probabilistic data association (JPDA) [8,9], and the random finite set method (RFS) [10–12]. The nearest neighbor association algorithm selects the observation point that falls within the association gate and is closest to the tracking target as the association point. The global nearest neighbor algorithm minimizes the total distance or association cost. The joint probabilistic data association algorithm combines data association probabilities. The multi-hypothesis tracking algorithm calculates the probability and likelihood for each track. The RFS approach models objects and measurements as random finite sets.

In high-resolution millimeter-wave radar or 4D millimeter-wave automotive radar, a road target often spans multiple sensor resolution units, which poses challenges for tracking. In the extended target tracking problem for millimeter-wave radar, the position of the target measurement point on the object is represented as a probability distribution that changes with the sensor measurement angle, and the measurement point may appear or disappear. Therefore, tracking extended targets using millimeter-wave radar presents a significant challenge.

One approach to address the extended target tracking problem is to include a clustering process that reduces multiple measurements to a single measurement, which can then be tracked using a point target tracking method. In extended target tracking, clustering can be used to partition the point cloud. In automotive millimeter-wave radar target tracking, the size and shape of the clustering clusters vary due to the different size and reflection properties of the targets. Therefore, density-based spatial clustering of applications with noise (DBSCAN) [13] is commonly used to cluster radar points. However, density-based clustering methods rely on fixed parameter values and may perform poorly with targets of different densities. As a result, several methods that allow for different clustering parameters have been proposed, such as ordering points to identify the clustering structure (OPTICS) [14], hierarchical DBSCAN (HSBSCAN) [15], and tracking-assisted multi-hypothesis clustering [16].

Other approaches to extended target tracking involve designing object measurement models. Some examples include the elliptic random matrix model [17], the random hypersurface model [18], and the Gaussian process model [19]. In millimeter-wave radar vehicle target tracking, various vehicle target models have been proposed, such as a direct

scattering model [20], a variational radar model [21], a B-spline chained ellipses model [22], and the data-region association model [23].

Although several methods exist for extended target tracking using millimeter-wave radar, many of them rely on simulation data for extended target tracking theory. In practical scenarios, challenges such as varied point cloud probability distributions of different extended targets, and diverse position relationships when different targets are associated require further investigation on certain tracking algorithms. Moreover, some algorithms focus on tracking vehicle targets, and thus it is essential to explore ways to adapt tracking algorithms to different types of targets with varying sizes. Additionally, there have been limited studies on 4D millimeter-wave radar target tracking, and, therefore, the effectiveness of such methods on 4D millimeter-wave radar needs to be explored. This paper presents an effective 4D millimeter-wave radar target tracking method with the following contributions.

1. This paper proposes a 4D millimeter-wave radar point cloud-based multi-target tracking algorithm for estimating the ID, position, velocity, and shape information of targets in continuous time.
2. The proposed target tracking solution includes point cloud velocity compensation, clustering, dynamic and static attribute update, dynamic target 3D border generation, static target contour update, and target trajectory management processes.
3. To address the issue of the varying size and shape of dynamic and static targets, a binary Bayesian filtering method [24] is utilized to extract static and dynamic targets during the tracking process.
4. Kalman filtering is used for dynamic targets such as vehicles, pedestrians, bicycles, and other targets, combined with the target's track information and radial velocity information to estimate the target's 3D border information.
5. For static targets such as road edges, green belts, buildings, and other non-regular shaped targets, the rolling ball method is employed to estimate and update the shape contour boundaries of the targets.

The structure of this paper is organized as follows. Section 2 describes the tracking problem. Section 3 presents the proposed solution to the tracking problem, which includes compensating for target velocity, clustering point clouds, determining target associations, identifying dynamic and static targets, updating contour shape states, and creating, retaining, and deleting trajectories. Section 4 presents the experimental setup and results. Finally, Section 5 summarizes the research.

## 2. Materials and Methods

The objective of this paper is to derive state estimates for both dynamic and static targets within the field of view of 4D millimeter-wave radar, using the point cloud measurement volume of the radar. This includes obtaining 3D edge information of dynamic targets and contour shape information of static targets.

### 2.1. Measurement Modeling

4D millimeter-wave radar point cloud measurement includes information on the position along the *x*, *y*, and *z*-axes as well as the radial velocity $v^r$ and intensity $I$. The radial velocity information is obtained through direct measurement as the target point's relative radial velocity. Each measurement point can be expressed as:

$$z_j = \begin{bmatrix} x_j & y_j & z_j & v_j^r & I_j \end{bmatrix} \tag{1}$$

where $z_j$ represents the measurement, $j$ represents the $j$-th point, and $v_j^r$ represents the relative radial velocity of the $j$-th point.

As shown in Figure 1, 4D millimeter-wave radar point clouds are utilized to measure targets at three distinct time steps, revealing that the detected target points are dynamic and can vary over time, possibly appearing or disappearing at different locations. This poses a

significant challenge in accurately estimating the target's location and shape. To account for sensor noise and the inherent uncertainty in the measurement model, a probabilistic model is often employed to describe the measurement process.

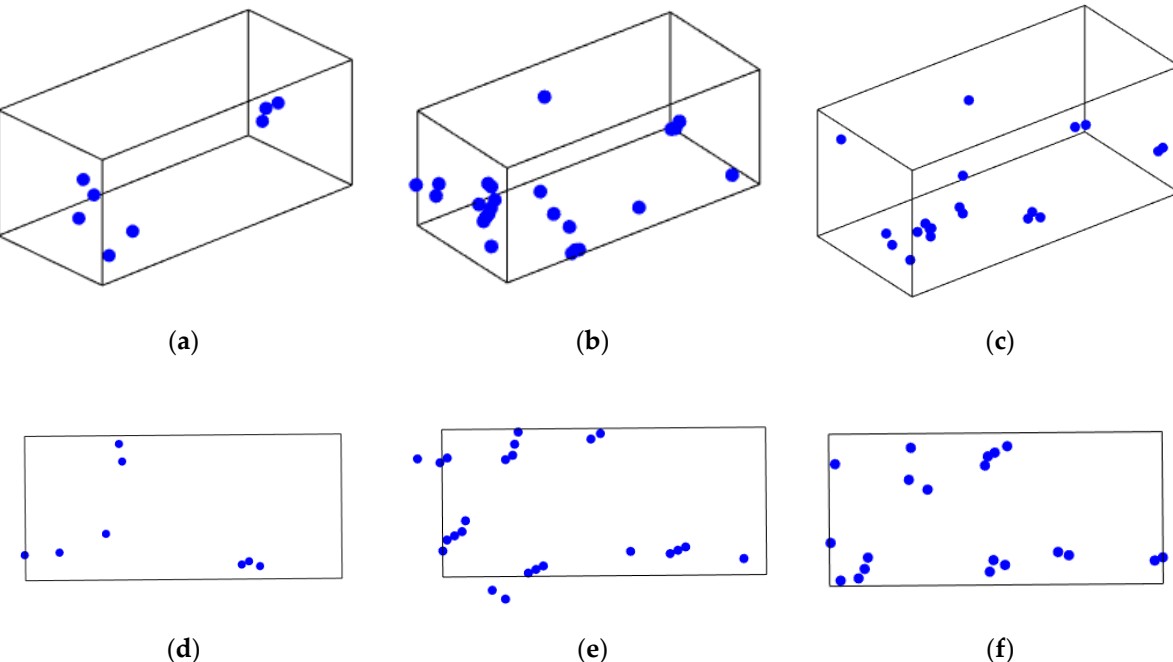

**Figure 1.** Measurements of the same target at adjacent moments. (**a**) 3D view of the target point cloud at moment $t-2$. (**b**) 3D view of the target point cloud at moment $t-1$. (**c**) 3D view of the target point cloud at moment $t$. (**d**) Top view of the target point cloud at moment $t-2$. (**e**) Top view of the target point cloud at moment $t-1$. (**f**) Top view of the target point cloud at moment $t$.

For multiple measurements of the expansion target, this can be expressed as:

$$Z = \left\{ z^j \right\}_{j=1}^{n} \tag{2}$$

where Z is the set of measurement quantities, $z^j$ is a single measurement quantity, $j$ is the number of measurements, and $n$ is the total number of measurements.

The probability distribution of the measurements obtained from the target state can be expressed as:

$$p(Z_k | X_k) \tag{3}$$

where $Z_k$ is the measurement at moment $k$ for a target with target state $X_k$.

*2.2. Target State Modeling*

The aim of this paper is to estimate the states of both dynamic and static targets in the 4D millimeter-wave radar field of view using point cloud measurements. For the dynamic targets, their states can be described as follows:

- Position state: The target's position in three-dimensional space ($x\ y\ z$).
- Motion state: Since the target's position in the z-axis direction remains relatively stable in autonomous driving scenarios, the motion state can be simplified to the target's velocity in the x-axis and y-axis directions on the vehicle motion plane ($v_x\ v_y$).
- Profile shape state: This describes the shape and size of the target. For a 3D dynamic target in a road environment, it can be modeled as a 3D cube ($l\ w\ h\ \theta$) since its shape and size states do not change substantially. Its extended state contains the size and rotation direction of the target.

Therefore, the state estimation of a 3D dynamic target in a road environment at time $k$ can be represented as $X_k$, which consists of the position state ($x$ $y$ $z$), the motion state ($v_x$ $v_y$), and the profile shape state ($l$ $w$ $h$ $\theta$).

$$X_k^d = [x_k \, y_k \, z_k \, v_{xk} \, v_{yk} \, l_k \, w_k \, h_k \, \theta_k] \tag{4}$$

The states of the static targets in this paper can be described as follows:

- Position state: The position of the target in the z-axis direction in space (z position).
- Motion state: For static targets, the absolute velocity is zero, and the relative velocity can be estimated as the negative of the velocity of the ego vehicle's motion ($v_{xk}$ $v_{yk}$).
- The profile shape state of the target: For a 3D static target in a road environment, it can be modeled as a target surrounded by an edge box, which is represented as a set of n 2D enclosing points and their heights ($h \left\{x_j \quad y_j\right\}_{j=1}^{n}$).

The state estimation of a 3D static target in a road environment can be expressed as:

$$X_d^s = [\{x_j \quad y_j\}_{j=1}^{n} \, v_{xk} \, v_{yk} \, h_k \, z_k] \tag{5}$$

### 2.3. Method

The proposed solution in this paper is illustrated in Figures 2 and 3:

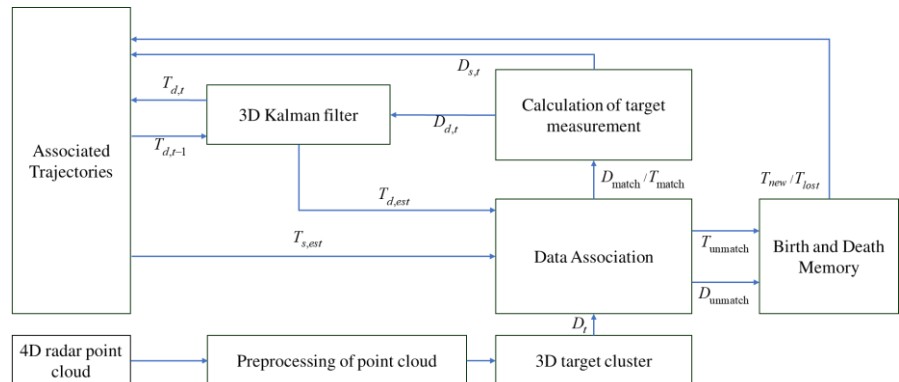

**Figure 2.** 4D millimeter-wave radar point cloud tracking framework.

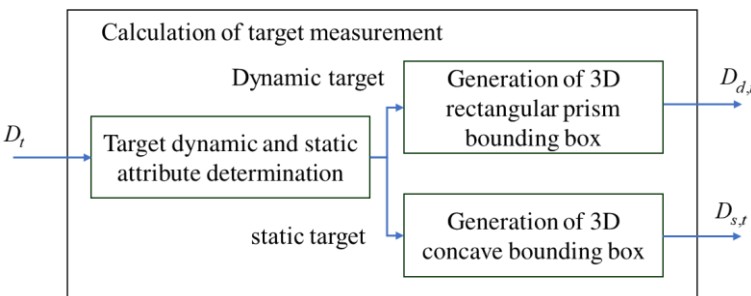

**Figure 3.** Module of calculation of target measurement.

In Figures 2 and 3, time is represented by $t$, the detection value is represented by $D$, the trajectory is represented by $T$, the dynamic target is represented by $d$, and the static target is represented by $s$.

The 4D radar data is input to generate point cloud data of the scene. The point cloud is preprocessed to compensate for the velocity information and convert relative radial velocity to absolute radial velocity. The static scene from the previous frame is matched with the current frame to aid in associating static and dynamic targets. A clustering module is used to classify the points into different target proposals. Data association is performed using an optimal matching algorithm. For the clustered targets that are successfully associated,

their dynamic and static attributes are updated using a binary Bayesian filtering algorithm. For dynamic targets, the target state is updated using a Kalman filtering method to obtain the 3D bounding box of the target. For static targets, the bounding box state is updated using the rolling ball method. For unassociated clustered targets, trajectory initialization is performed, historical trajectories that are not associated are retained or deleted, and trajectories in overlapping regions are merged.

### 2.3.1. Point Cloud Preprocessing

Before feeding the millimeter-wave radar point cloud into the tracking framework, several preprocessing steps are performed. Firstly, the relative radial velocity information of the point cloud is compensated for absolute radial velocity, allowing for the extraction of dynamic and static targets in the scene and the updating of their states based on radial velocity information. Additionally, due to the motion of the radar, the world coordinate systems of the front and back point clouds are different, and multi-frame point clouds are matched to facilitate the association of dynamic and static targets. Further details on these steps can be found in previous work [25].

After obtaining the ego vehicle's speed $v_s$, the compensation amount, $\hat{v}_c^r$, for the radial velocity of the target can be calculated. Then, the absolute velocity of each target point, $v_a^r$, can be calculated as follows:

$$v_a^r = v_d^r - \hat{v}_c^r \tag{6}$$

The radar point cloud conversion relationship can be expressed as:

$$H = [R, t] \tag{7}$$

$$Y_{n-1}^n = H_{n-1} P_{n-1} \tag{8}$$

$Y_{n-1}^n$ is the point set after the point cloud of the $(n-1)$-th frame is registered to the point cloud of the $n$-th frame. $P_{n-1}$ is the information of the $n$-th point.

### 2.3.2. Clustering and Data Association

- Radar Point Cloud Clustering

After preprocessing the point cloud data, the large number of points are grouped into different targets using clustering techniques based on their position and velocity characteristics. One commonly used clustering algorithm for radar point clouds is density-based spatial clustering of applications with noise (DBSCAN) [13], which can automatically detect clustering structures of arbitrary shapes without requiring any prior knowledge. DBSCAN determines clusters by calculating the density around sample points, grouping points with higher density together to form a cluster, and determining the boundary between different clusters by the change in density. The DBSCAN algorithm takes spatial coordinates $(x, y, z)$ and radial distance $(v_a^r)$ of the data points as input. Specifically, the DBSCAN algorithm can be executed in the following steps:

- Calculation of the number of data points $N(p)$ in the neighborhood of a data point $p$:

$$N(p) = \{q \in Z : dist(p, q) \leq \varepsilon\} \tag{9}$$

Here, $Z$ is the dataset, $dist(p, q)$ is the Euclidean distance between the data points $p$ and $q$, and $\varepsilon$ is the radius of the neighborhood.

- Determination of whether a data point $p$ is a core point: If $N(p) \geq MinPts$, then $p$ is a core point.
- Expanding the cluster: Starting from any unvisited core point, find all data points that are density-reachable from the core point, and mark them as belonging to the same cluster.

- Determination of whether a data point is density-reachable: A data point $p$ is density-reachable from a data point $q$ if there exists a core point $c$ such that both $c$ and $p$ are in the neighborhood of $q$ and the distance between $c$ and $p$ is less than $\varepsilon$.
- Marking noise points: Any unassigned data points are marked as noise points.

By executing the above steps, the DBSCAN algorithm can complete the clustering process and assign the data points to different clusters and noise points.

After clustering the $k$ targets, the features of the $j$-th target are represented as:

$$f_j = \left\{ \overline{x}_j \quad \overline{y}_j \quad \overline{z}_j \quad \overline{v}_j^r \quad \overline{I}_j \right\} \tag{10}$$

where ( $\overline{x}_j \quad \overline{y}_j \quad \overline{z}_j \quad \overline{v}_j^r \quad \overline{I}_j$ ) are calculated as the averages of the point cloud features within each target. The features of all clustering targets can be expressed as:

$$F = \left\{ f_j \right\}_{j=1}^n \tag{11}$$

- Data Association

For the $j$-th trajectory, its features are denoted as:

$$g_j = \left\{ \widetilde{x}_j \quad \widetilde{y}_j \quad \widetilde{z}_j \quad \widetilde{v}_j \quad \widetilde{I}_j \right\} \tag{12}$$

The features of all trajectories can be expressed as:

$$G = \left\{ g_j \right\}_{j=1}^n \tag{13}$$

The purpose of data correlation is to select which measurements are used to update the state estimate of the real target and to determine which measurements come from the target and which come from clutter. In this paper, it is necessary to correlate all clustered targets $F$ and all trajectories $G$. One of the most widely used algorithms for target association is the Hungarian algorithm, which is a classical graph theoretic algorithm that can be used to maximize the matching of bipartite graphs. It can be used in a variety of target association algorithms for radar or images, and in target tracking it can be used to match point clouds in target clusters at different time steps to achieve target association. Assuming that there are radar historical trajectories and clustered targets, where the clustered targets contain m targets and the radar trajectories contain n targets, a cost matrix can be defined where $\text{Cost}(i,j)$ denotes the cost between the $i$-th point in the trajectory and the $j$-th point in the clustered targets. Depending on the needs of the target tracking, the cost function can be calculated from factors such as target clustering centroids, average velocity, and intensity characteristics. The Hungarian algorithm finds the optimal matching solution with the minimum cost by converting the bipartite graph into a directed complete graph with weights and by finding the augmented paths in the graph.

The substitution matrix is calculated using the cost function, which is a combination of the position cost and the velocity/intensity cost. The cost function is defined as:

$$\text{Cost}(i,j) = \alpha_1 \times PositionCost(i,j) + \alpha_2 \times VelocityIntensityCost(i,j)) \tag{14}$$

where $\alpha_1$ is the weight of the position cost and $\alpha_2$ is the weight of the velocity/intensity cost. The position cost can be calculated based on the distance between the target centroid and the trajectory prediction at the current time step, while the velocity/intensity cost can be calculated based on the difference in velocity and intensity between the target and the trajectory prediction.

Once the cost function has been calculated, the Hungarian algorithm can be used to find the optimal matching solution with the minimum cost. The resulting substitution matrix $C$ is a binary matrix, where $C(i,j) = 1$ if target $i$ is matched to the trajectory $j$, and $C(i,j) = 0$ otherwise.

### 2.3.3. Target Status Update

- Target Dynamic Static Property Update

By integrating the absolute velocity information of a target with a binary Bayesian filter, its static and dynamic attributes can be updated. To estimate the target's dynamic probability at a given moment, the ratio of points with a speed greater than a given value to the total number of points in the target's point cloud is calculated. Bayes' theorem is used in the binary Bayesian filter to update the state of the target, which can be either static or dynamic, represented by a binary value of 0 or 1, respectively, at time t.

Applying Bayes' theorem:

$$p(x|z_{1:t}) = \frac{p(z_t|x, z_{1:t-1})p(x|z_{1:t-1})}{p(z_t|z_{1:t-1})} = \frac{p(z_t|x)p(x|z_{1:t-1})}{p(z_t|z_{1:t-1})} \tag{15}$$

The Bayes' rule is applied to the measurement mode $p(z_t|x)$:

$$p(z_t|x) = \frac{p(x|z_t)p(z_t)}{p(x)} \tag{16}$$

Then,

$$p(x|z_{1:t}) = \frac{p(x|z_t)p(z_t)p(x|z_{1:t-1})}{p(x)p(z_t|z_{1:t-1})} \tag{17}$$

For the opposite event $\neg x$,

$$p(\neg x|z_{1:t}) = \frac{p(\neg x|z_t)p(z_t)p(\neg x|z_{1:t-1})}{p(\neg x)p(z_t|z_{1:t-1})} \tag{18}$$

Then,

$$\frac{p(x|z_{1:t})}{p(\neg x|z_{1:t})} = \frac{p(x|z_t)p(x|z_{1:t-1})p(\neg x)}{p(\neg x|z_t)p(\neg x|z_{1:t-1})p(x)} = \frac{p(x|z_t)}{1 - p(x|z_t)} \frac{p(x|z_{1:t-1})}{1 - p(x|z_{1:t-1})} \frac{1 - p(x)}{p(x)} \tag{19}$$

The log odds belief at time $t$ is:

$$l_t(x) = \log \frac{p(x|z_t)}{1 - p(x|z_t)} - \log \frac{p(x)}{1 - p(x)} + l_{t-1}(x) \tag{20}$$

And,

$$l_0(x) = \log \frac{p(x)}{1 - p(x)} \tag{21}$$

Then,

$$l_t(x) = l_{t-1}(x) + \log \frac{p(x|z_t)}{1 - p(x|z_t)} - l_0 \tag{22}$$

In dynamic and static attribute updates, $p(x|z_t)$ is calculated as the ratio of the number of points with a velocity greater than a given value $v_d$ to the total number of points in the target point cloud.

- Dynamic Target State Update

The state estimation of a 3D dynamic target in a road environment at time $k$ can be represented as $X_k^d$ by Equation (4), which consists of the position state $(x\ y\ z)$, the motion state $(v_x\ v_y)$, and the profile shape state $(l\ w\ h\ \theta)$.

To update the state of a target, it is necessary to perform additional calculations on the existing clustered targets to obtain measurements of its current state. These calculations may involve analyzing the shape and center position of the target, as well as estimating its velocity. Once these calculations are completed, the status of the target can be updated based on the latest information available, allowing for more accurate tracking and prediction of the target's movement.

When computing measurements of clustered targets for dynamic targets, it is necessary to obtain the rectangular box enclosing the target. The height of the rectangular box can be calculated from the maximum and minimum height of the point cloud, while the other parameters of the rectangular box can be obtained from the enclosing rectangular box in the x and y planes.

However, calculating the rotation angle of the rectangular box is the most challenging part of target shape estimation, especially in imaging millimeter-wave radar, where the number of point clouds is limited and the contours of the point clouds are not well-defined. To address this issue, this paper proposes a method for calculating the rotation angle based on the combination of point cloud position and velocity information and trajectory angle. This approach provides a more accurate and robust estimate of the rotation angle, leading to improved target tracking and prediction.

The rectangular box of the point cloud is fitted using the L shape fitting method [26]. When working with points on a 2D plane, the least squares method is a common approach to finding the best-fitting rectangle for these points.

$$\begin{aligned} \underset{P,\theta,c_1,c_2}{\text{minimize}} \sum_{i \in P} (x_i \cos\theta + y_i \sin\theta - c_1)^2 + \sum_{i \in Q} (-x_i \sin\theta + y_i \cos\theta - c_2)^2 \\ \text{subject to } P \cup Q = \{1, 2, \ldots, m\} \; c_1, c_2 \in R \; 0° \leq \theta \leq 90° \end{aligned} \tag{23}$$

The above optimization problem can be approximated by using a search-based algorithm to find the best-fitting rectangle. The basic idea is to iterate through all possible directions of the rectangle. At each iteration, a rectangle is found that points in that direction and contains all scanned points. The distances from all points to the four edges of the rectangle are then obtained, based on which the points can be divided into two sets, p and q, and the corresponding squared errors are calculated as the objective function in the above equation. After iterating through all directions and obtaining all corresponding squared errors, the squared errors can be plotted as a function of the angle variation trend. Algorithm 1 is as follows.

---

**Algorithm 1**

---

- **Input**: data points $X = (x, y)$
- **Output**: criterion $Q_p$

1.     **For** $\theta = 0$ **to** $\pi/2 - \delta$ step $\delta$ do
2.         $\hat{e}_1 = (\cos\theta, \sin\theta)$
3.         $\hat{e}_2 = (-\sin\theta, \cos\theta)$
4.         $C_1 = X \cdot \hat{e}_1^T$
5.         $C_2 = X \cdot \hat{e}_2^T$
6.         $q = \text{CalculatecriterionX}(C_1, C_2)$
7.         $Q_p(\theta) = q$
8.     end for

---

The algorithm for defining the calculate criterion, $\text{CalculatecriterionX}(C_1, C_2)$, using the minimum rectangular area method as described in this paper, is as follows:

$$c_1^{\max} = \max\{C_1\}, \; c_1^{\min} = \min\{C_1\} \tag{24}$$

$$c_2^{\max} = \max\{C_1\}, \; c_1^{\min} = \min\{C_2\} \tag{25}$$

$$\alpha = -(c_1^{\max} - c_1^{\min})(c_2^{\max} - c_2^{\min}) \tag{26}$$

After calculating to obtain $Q_p(\theta)$, the probability $P_p(\theta)$ is calculated as:

$$P_p(\theta) = \frac{\max(Q(\theta)) - Q_p(\theta) + \min(Q_p(\theta))}{\sum\limits_{\theta} Q_p(\theta)} \tag{27}$$

For a target on a two-dimensional plane, if the velocities of the point clouds on the target are assumed to be approximately equal, the orientation of the velocities can be estimated. Since millimeter-wave radar has different radial velocities at different points, this estimated velocity orientation can be used as an approximation for the rotation angle of the estimated rectangle for the calculation of the rotation angle, as follows.

The radial velocity measured by millimeter-wave radar can be expressed as

$$v_d^r = v_{d,x}\frac{x}{R} + v_{d,y}\frac{y}{R} \tag{28}$$

$$v_d^r = v_{d,x}(\frac{x}{R} + \tan\theta\frac{y}{R}) \tag{29}$$

Similar can be achieved by using a search-based algorithm to find the right angle, where the criterion is calculated as the variance. Algorithm 2 is as follows.

---

**Algorithm 2**

---

- **Input**: $X = (\frac{x}{Rv_d^r}, \frac{y}{Rv_d^r})$
- **Output**: criterion $Q_v$

1.  **For** $\theta = 0$ **to** $2 * \pi - \delta$ step $\delta$ do
2.      $\hat{e} = (1, \tan\theta)$
3.      $C = X \cdot \hat{e}^T$
4.      $q = \text{variance}\{C\}$
5.      $Q_v(\theta) = q$
6.  **end for**

---

After calculating to obtain $Q_v(\theta)$, the probability $P_v(\theta)$ is calculated as:

$$P_v(\theta) = \frac{\max(Q_v(\theta)) - Q_v(\theta) + \min(Q_v(\theta))}{\sum\limits_{\theta} Q_v(\theta)} \tag{30}$$

Calculating the historical trajectory angle as $\theta_l$ and the probability as a Gaussian distribution with center at $\theta_l$ and variance at $\delta_l$:

$$P_t(\theta) = N(\theta, \sigma^2) + P_t \tag{31}$$

$$P_h(\theta) = \frac{P_t(\theta)}{\sum\limits_{\theta} P_t(\theta)} \tag{32}$$

Angular probabilities estimated from the point cloud position and velocity information and trajectory angles are fused using a weighted average.

$$P(\theta) = \alpha_1 P_p(\theta) + \alpha_2 P_v(\theta) + \alpha_1 P_h(\theta) \tag{33}$$

The theta value that maximizes $P(\theta)$ is chosen as the measured value, and the rectangular boundary $\{a_i x + b_i y = c_i | i = 1, 2, 3, 4\}$ is calculated as:

$$C_1^* = X \cdot (\cos\theta^*, \sin\theta^*)^T, C_2^* = X \cdot (-\sin\theta^*, \cos\theta^*)^T \tag{34}$$

$$a_1 = \cos\theta^*, b_1 = \sin\theta^*, c_1 = \min\{C_1^*\} \tag{35}$$

$$a_2 = -\sin\theta^*, b_2 = \cos\theta^*, c_2 = \min\{C_2^*\} \tag{36}$$

$$a_3 = \cos\theta^*, b_3 = \sin\theta^*, c_3 = \max\{C_1^*\} \tag{37}$$

$$a_4 = -\sin\theta^*, b_4 = \cos\theta^*, c_4 = \max\{C_2^*\} \tag{38}$$

From the process described above, the following parameters of the clustered target can be calculated: the centroid coordinates in three-dimensional space $(x, y, z)$, the length, width, and height of the rectangular box enclosing the target, and the rotation angle $(\theta)$ of the rectangular box.

The velocity information of the target can be calculated by Equation (32).

Then, the measurement can be expressed as:

$$Z_{t,k} = \begin{bmatrix} x_k & y_k & z_k & v_{xk} & v_{yk} & l_k & w_k & h_k & \theta_k \end{bmatrix} \tag{39}$$

The state transfer model of the target motion can be modeled as:

$$X_t = FX_{t-1} + \xi_t \tag{40}$$

where $\xi_t$ is the system white Gaussian noise with covariance $\eta(\xi; 0, R)$.

The sensor's observation model is described as:

$$z_t = Hx_{t-1} + \zeta_t \tag{41}$$

where $\zeta_t$ is the measurement white Gaussian noise with covariance $\eta(\zeta; 0, Q)$.

Based on Equations (40) and (41), since the state and measurement equations of the target can be expressed in linear forms, the state can be updated by the Kalman filter.

- Static Target State Update

The state estimation of a 3D static target in a road environment can be expressed as Equation (5)

When calculating measurements for clustered target detection in static scenarios, obtaining the enclosing box of the target is necessary. The height of the enclosing box can be determined by computing the maximum and minimum heights of the point cloud, while the other parameters of the enclosing box can be obtained from the enclosing concave hull in the x and y planes.

The specific steps of the algorithm are as follows:

1. For any point $p$ and rolling ball radius $a$, search for all points within a distance of $2a$ from $p$ in the point cloud, denoted as the set $Q$.
2. Select any point $p_1(x, y)$ from $Q$ and calculate the coordinates of the center of the circle passing through $p$ and $p_1$ with a radius of alpha. There are two possible center coordinates, denoted as $p_2$ and $p_3$.
3. Remove $p_1$ from the set $Q$ and calculate the distances between the remaining points and the points $p_2$ and $p_3$. If all distances are greater than $a$, the point $p$ is considered a boundary point.
4. If all distances are not greater than $a$, iterate over all points in $Q$ as the new $p$ and repeat steps (2) and (3). If a point is found that satisfies the conditions in steps (2) and (3), it is considered a boundary point and the algorithm moves on to the next point. If no such point is found among the neighbors of $p$, then $p$ is considered a non-boundary point.

Through Formula (7) of the radar point cloud velocity compensation part, $v_{xk}$ and $v_{yk}$ of the static target can be calculated, and the vehicle speed can be updated through the Kalman filter.

### 2.3.4. Track Management

In multi-object tracking, the number of targets is typically unknown and can vary as targets and clutter appear and disappear from the scene. Therefore, effective management of target trajectories is essential. For associated detections and trajectories, their states are preserved and updated over time. In cases where detections cannot be associated with any existing trajectory, new trajectories are generated and released as visible trajectories if their lifespan exceeds a predefined threshold $T_r$. For unassociated trajectories, their states are also preserved and updated. However, if their unassociated time exceeds a second threshold $T_u$, the trajectories are deleted to avoid unnecessary computational load.

## 3. Results

### 3.1. Experiment Setup

To verify the proposed algorithm, data from a 4D radar in road conditions were acquired using a data acquisition platform. The platform includes a 4D radar, LIDAR, and camera sensors, as shown in Figure 4. The 4D radar is installed in the middle of the front ventilation grille, and the LIDAR collects 360° of environmental information. The camera and 4D radar capture information within the field of view. The true value frame of the tracking target was labeled using the LIDAR and camera sensors. The performance parameters of the 4D radar sensor are shown in Table 1. The TJ4DRadSet [27] dataset was collected and is used for the algorithm analysis. As shown in Figure 4, the collection platforms of the dataset are displayed.

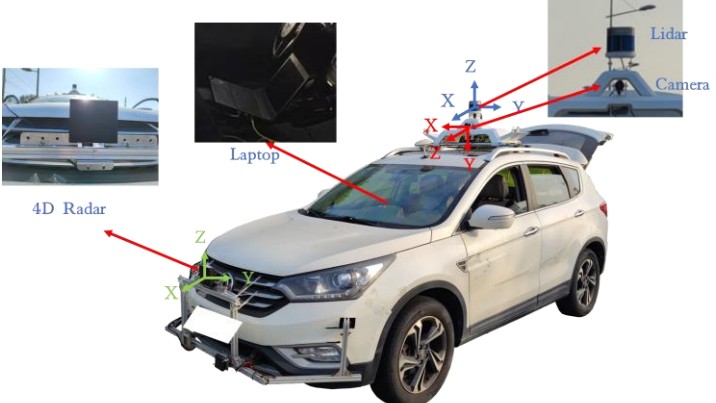

**Figure 4.** Data acquisition platform, including 4D radar, lidar, and camera sensor.

**Table 1.** Performance parameters of millimeter-wave radar in experimental data acquisition.

| Sensors | Resolution | | | FOV | | |
|---|---|---|---|---|---|---|
| | Range | Azimuth | Elevation | Range | Azimuth | Elevation |
| 4D radar | 0.86 m | <1° | <1° | 400 m | 113° | 45° |

### 3.2. Results and Evaluation

In order to investigate the impact of velocity errors on the angle estimation under different distances and angles, the graphs shown in Figures 5–10 were plotted.

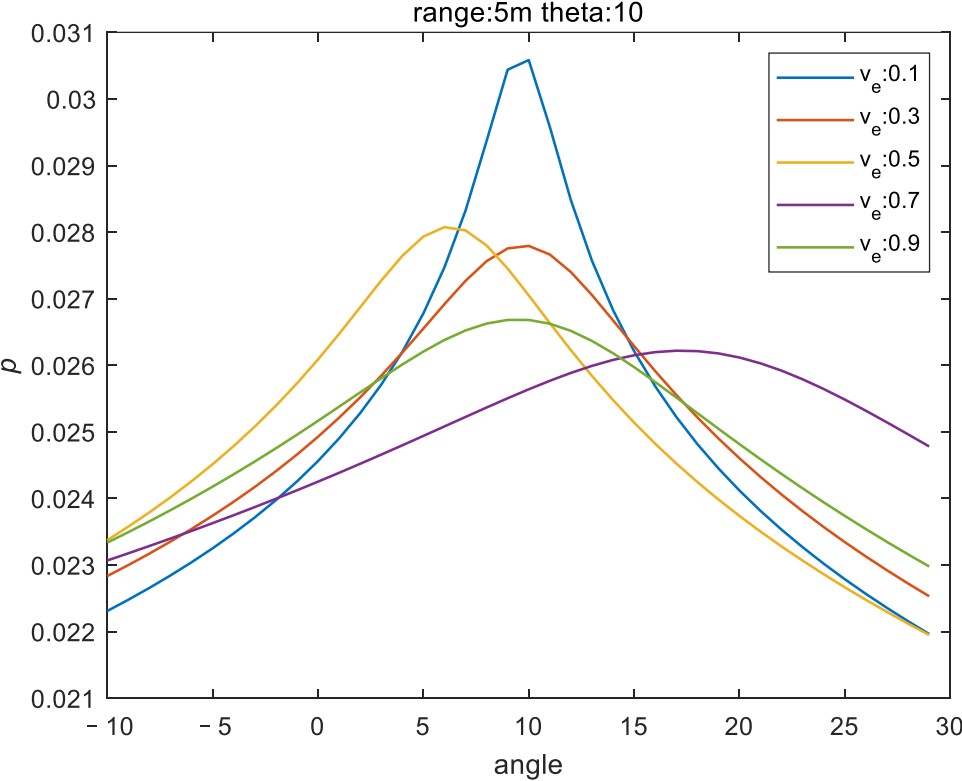

**Figure 5.** The relationship between velocity estimation and angle under a distance of 5 m and an object rotation angle of 10 degrees, considering different radial velocity measurement errors.

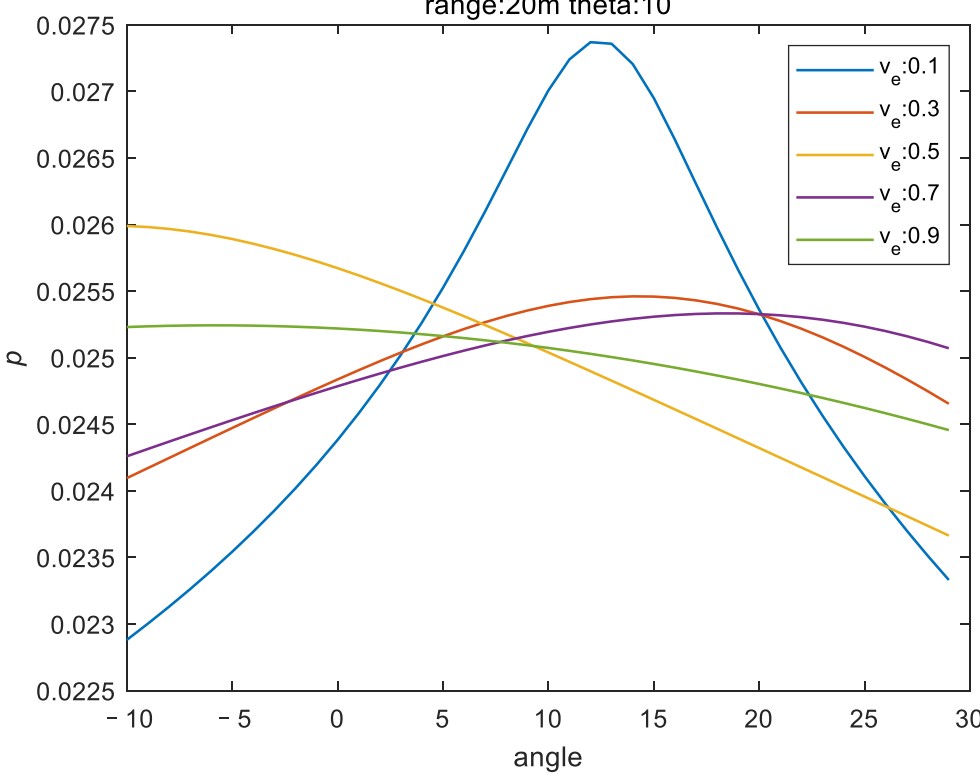

**Figure 6.** The relationship between velocity estimation and angle under a distance of 20 m and an object rotation angle of 10 degrees, considering different radial velocity measurement errors.

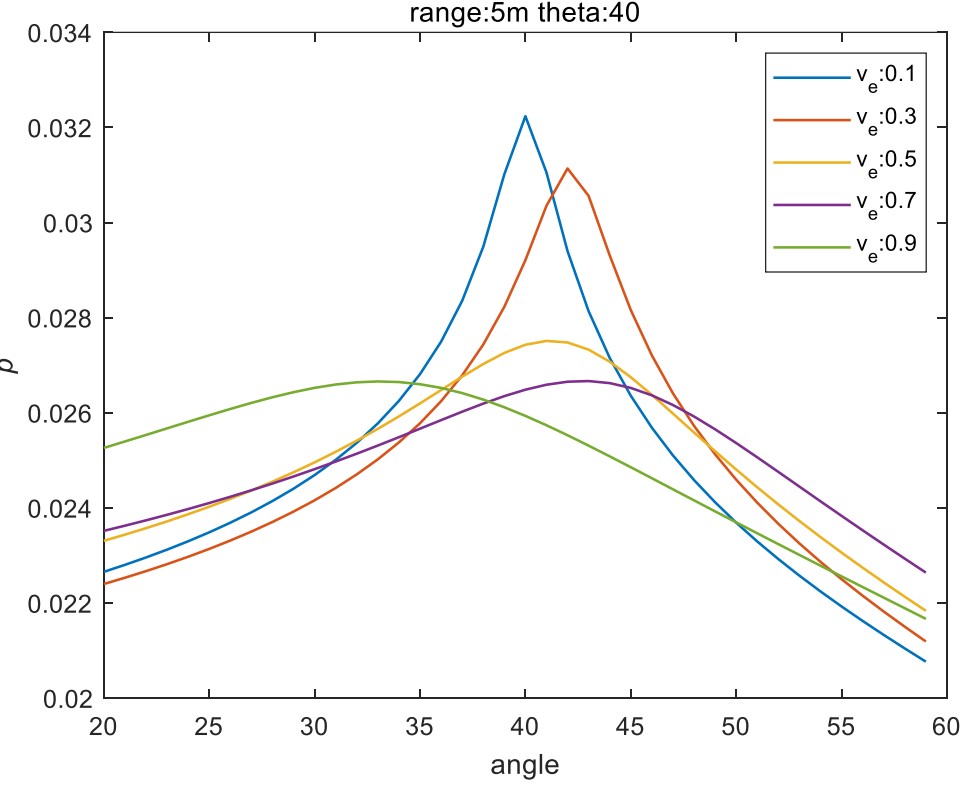

**Figure 7.** The relationship between velocity estimation and angle under a distance of 5 m and an object rotation angle of 40 degrees, considering different radial velocity measurement errors.

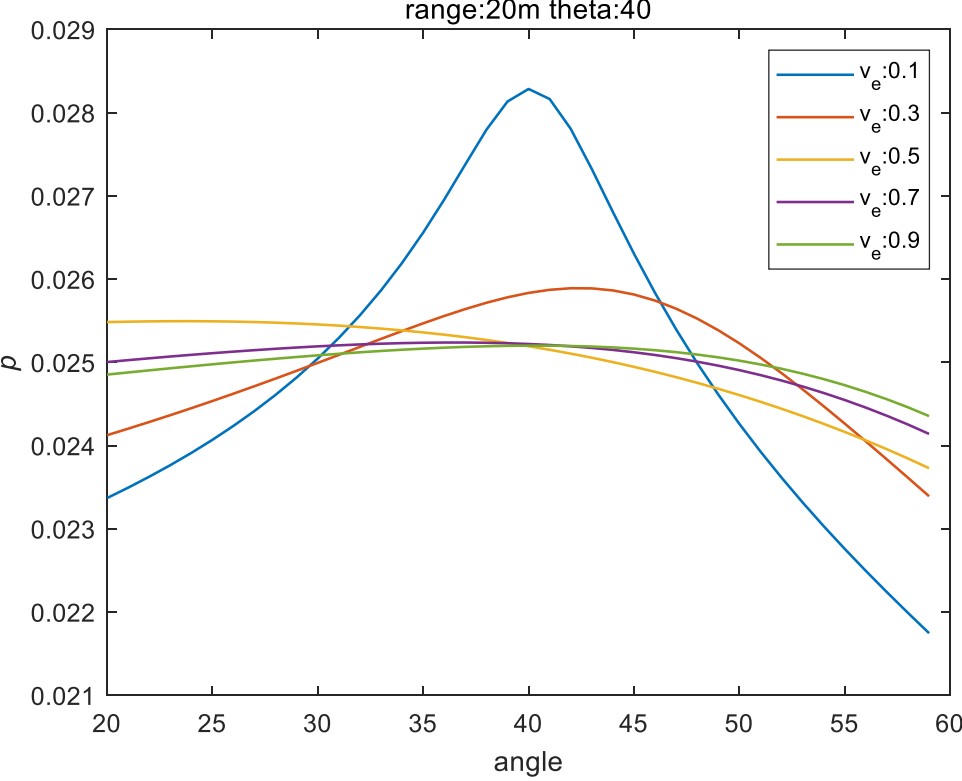

**Figure 8.** The relationship between velocity estimation and angle under a distance of 20 m and an object rotation angle of 40 degrees, considering different radial velocity measurement errors.

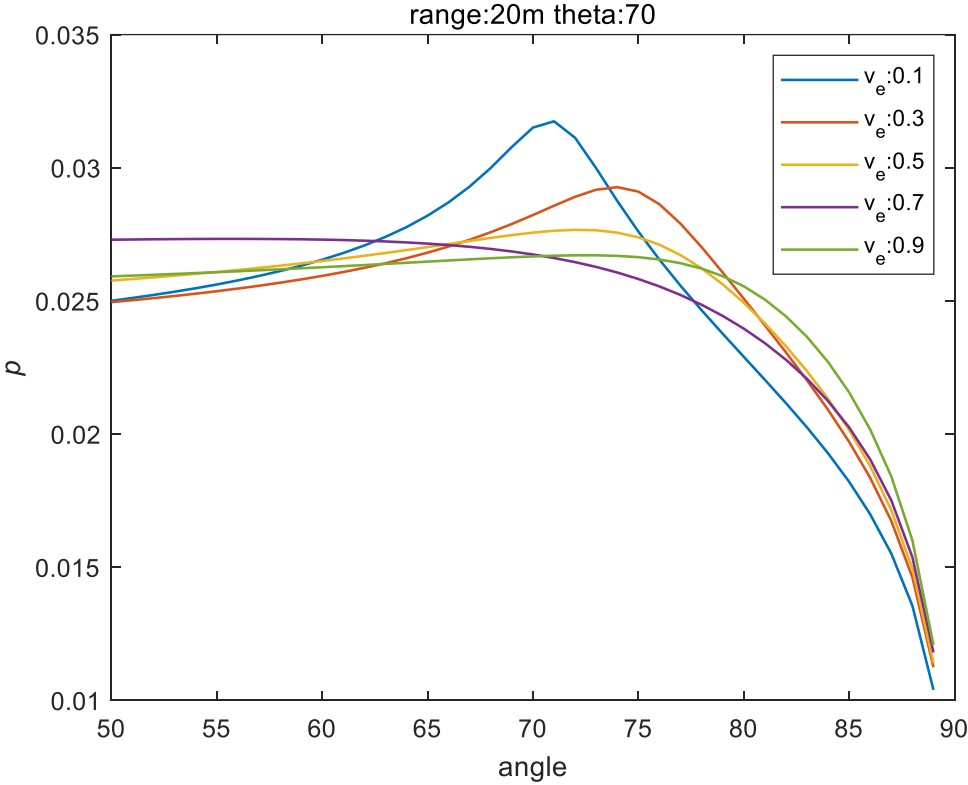

**Figure 9.** The relationship between velocity estimation and angle under a distance of 20 m and an object rotation angle of 70 degrees, considering different radial velocity measurement errors.

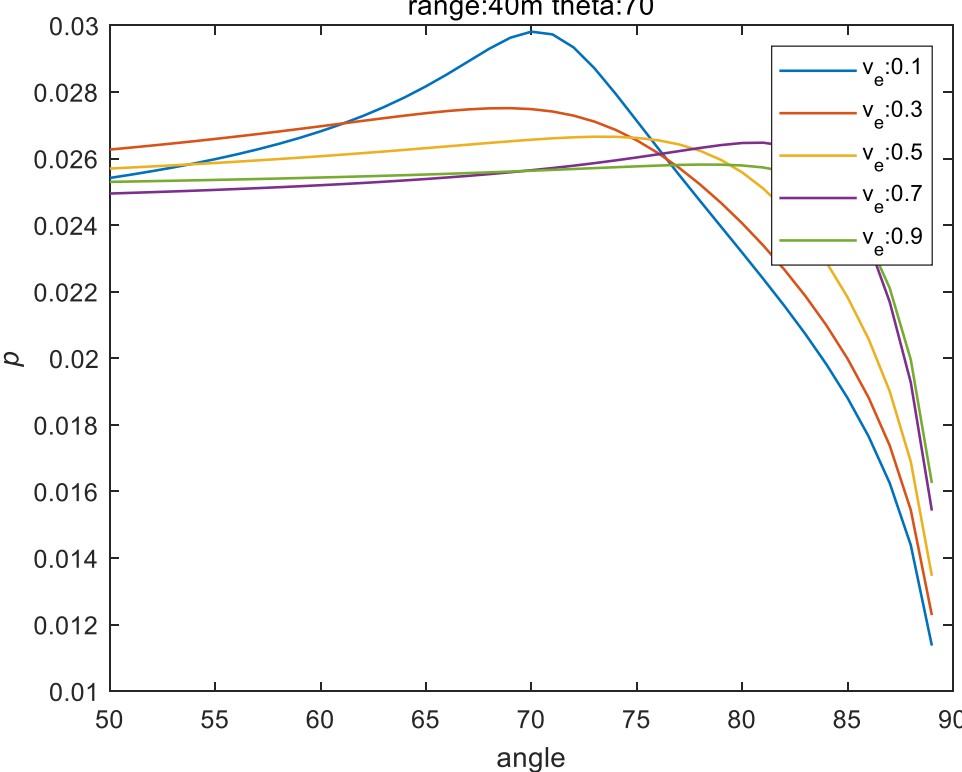

**Figure 10.** The relationship between velocity estimation and angle under a distance of 40 m and an object rotation angle of 70 degrees, considering different radial velocity measurement errors.

From Figures 5–10, it can be observed that when the radial velocity error is small, the estimation of the rotation angle can be made using velocity measurements from multiple points, and a shorter distance is more favorable for estimating the rotation angle based on the velocity.

Due to the limited number of millimeter-wave radar points, the rotation angle estimation of the dynamic target is fused by different methods. As shown in Figures 11 and 12, the rotation angle of the dynamic target can be better estimated.

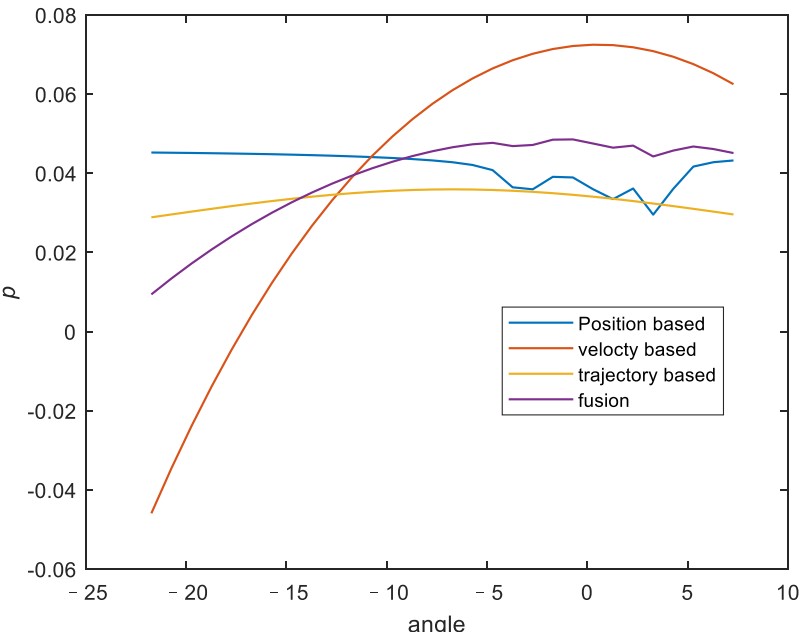

**Figure 11.** Method for estimating dynamic targets at different angles, and the relationship between probability and angle changes.

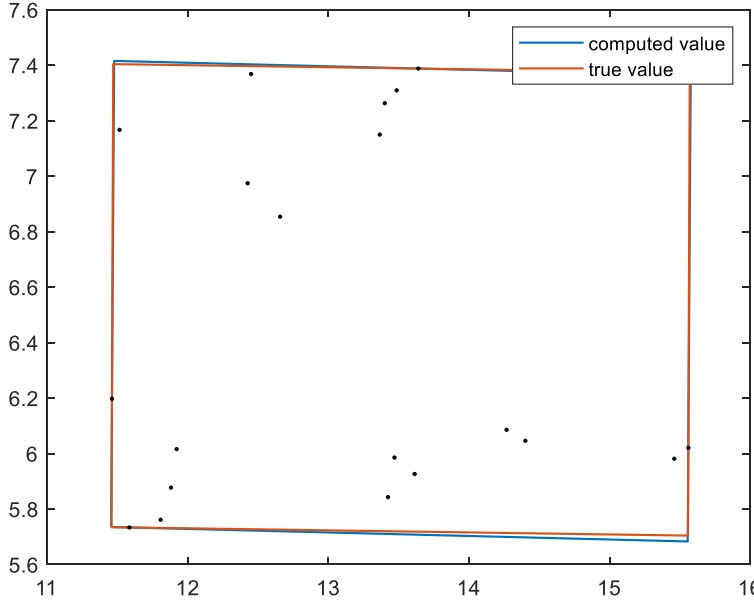

**Figure 12.** Rectangle formed by the estimated rotation angle and the true rotation angle.

Figures 13–15 show the state estimation of dynamic targets and static targets in a 4D millimeter-wave radar scenario. Different estimated dynamic targets, static targets, and true bounding boxes of dynamic targets have been labeled.

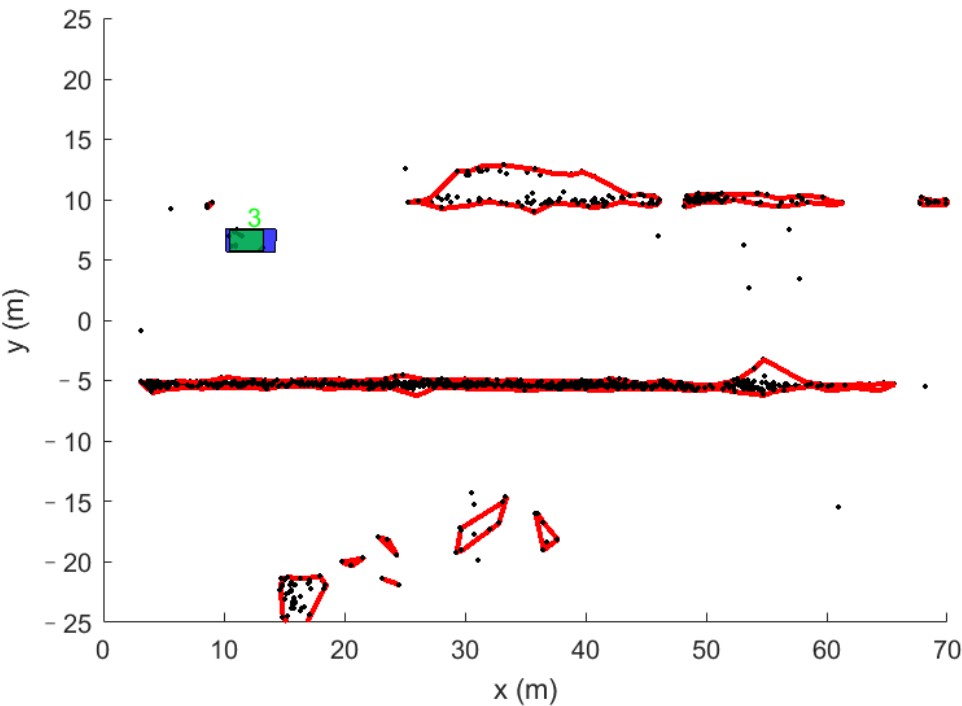

**Figure 13.** Results of 4D millimeter-wave radar point cloud and target tracking for a single vehicle, where the green box represents a dynamic target, the red box represents a static target, and the blue box represents the true box of a dynamic target.

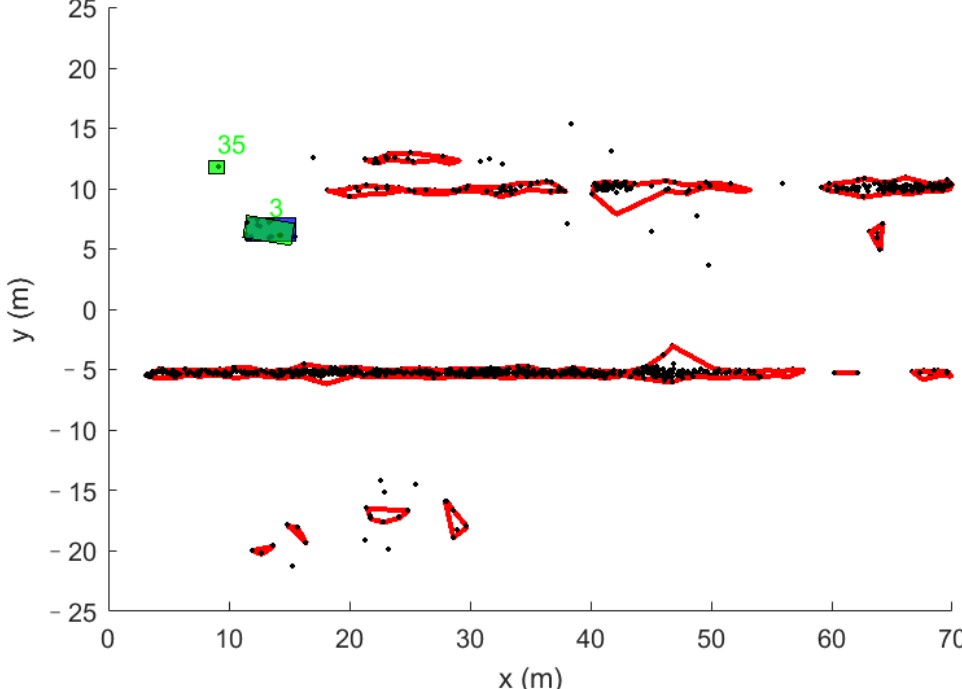

**Figure 14.** Results of 4D millimeter-wave radar point cloud and target tracking for a single vehicle, including an incorrect dynamic detection, where the green box represents a dynamic target, the red box represents a static target, and the blue box represents the true box of a dynamic target.

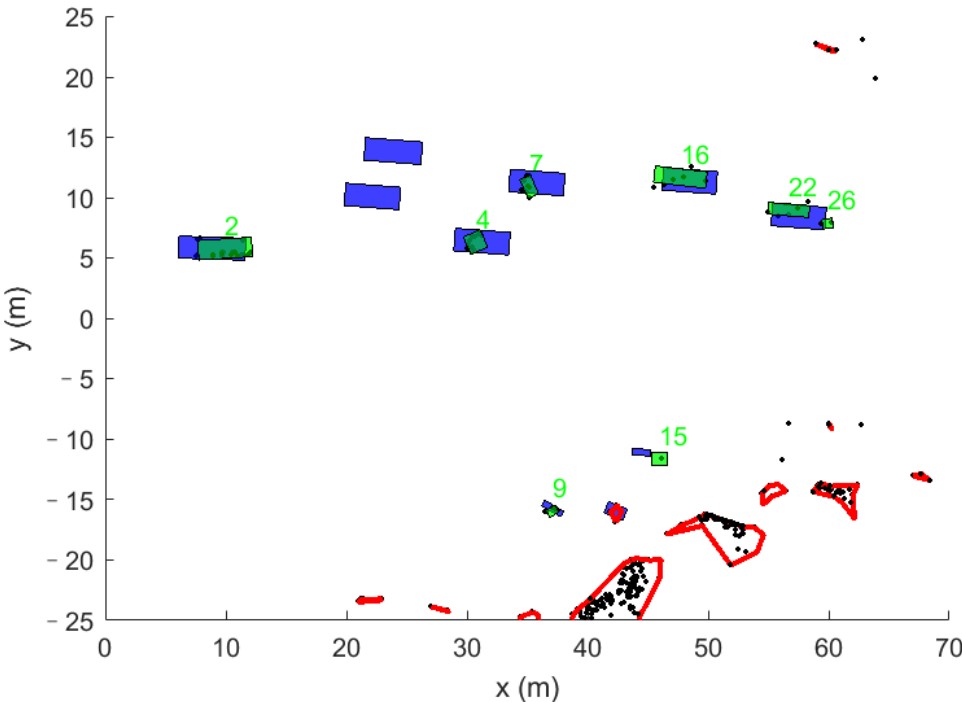

**Figure 15.** Results of 4D millimeter-wave radar point cloud and target tracking for multiple objects, where the green box represents a dynamic target, the red box represents a static target, and the blue box represents the true box of a dynamic target.

Figure 16 shows the effects of different performance parameters in the target tracking scene.

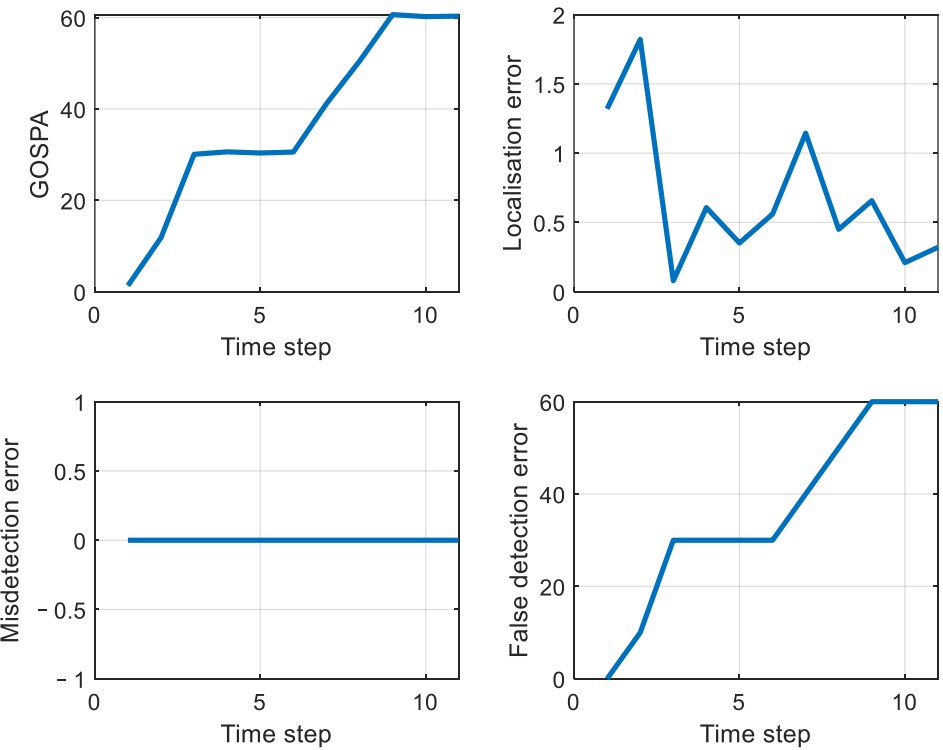

**Figure 16.** Performance curves of different indicators for dynamic targets in a tracking scenario.

## 4. Discussion

The proposed 4D radar object tracking method based on radar point clouds can effectively estimate the position and state information of radar targets. This provides more accurate information for perception and planning in autonomous driving. By utilizing radar point clouds, the method improves the tracking and prediction of surrounding objects, enabling autonomous vehicles to make informed decisions in real time. Precise localization and tracking of radar targets enhance situational awareness, allowing autonomous vehicles to navigate complex environments with greater reliability and safety. Overall, this method significantly enhances the perception and planning capabilities of autonomous driving systems, contributing to the development of safer and more efficient autonomous vehicles.

## 5. Conclusions

In summary, this paper presents a 4D radar-based target tracking algorithm framework that utilizes 4D millimeter-wave radar point cloud information for autonomous driving awareness applications. The algorithm overcomes the limitations of conventional 2 + 1D radar systems and utilizes higher resolution target point cloud information to achieve more accurate motion state estimation and target profile information. The proposed algorithm includes several steps, such as ego vehicle speed estimation, density-based clustering, and binary Bayesian filtering to identify dynamic and static targets, as well as state updates of dynamic and static targets. Experiments are conducted using measurements from 4D millimeter-wave radar in a real-world in-vehicle environment, and the algorithm's performance is validated by actual measurement data. The algorithm can improve the accuracy and reliability of target tracking in autonomous driving applications. This method focuses on the tracking framework for 4D radar. However, further research is needed to investigate the details of certain aspects such as motion models, filters, and ego-vehicle pose estimation.

**Author Contributions:** Conceptualization, B.T., Z.M. and X.Z.; methodology, B.T., Z.M. and X.Z.; software, B.T.; validation, B.T., S.L. and L.Z.; formal analysis, L.Z.; investigation, S.L.; resources, L.H.; data curation, B.T. and L.Z.; writing—original draft preparation, Z.M.; writing—review and editing, B.T. and Z.M.; visualization, L.Z.; supervision, X.Z. and L.H.; project administration, X.Z. and J.B.; funding acquisition, X.Z. and J.B. All authors have read and agreed to the published version of the manuscript.

**Funding:** This research was funded by the National Key R&D Program of China (2022YFB2503404).

**Data Availability Statement:** Not applicable.

**Conflicts of Interest:** The authors declare no conflict of interest.

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
