# Peer review of "Tracking of Multiple Static and Dynamic Targets for 4D Automotive Millimeter-Wave Radar Point Cloud in Urban Environments"

_remotesensing, doi:10.3390/rs15112923_

Round 1
Reviewer 1 Report
This paper proposes a novel method for autonomous vehicle target tracking. This paper is interesting and the result is convincing. Some minor modifications should be made before publication.
1) Kalman filters are a popular method to estimate autonomous vehicle states as they are easy to follow and robust. Currently, there are a lot of filter variants to optimize the state estimation. And the host vehicle velocity is pretty important for target status update. Thus, some related work should be included: autonomous vehicle kinematics and dynamics synthesis for sideslip angle estimation based on consensus kalman filter, automated vehicle sideslip angle estimation considering signal measurement characteristic.
2) Based on the sensors’ configuration of autonomous vehicles, there are some alternative methods for object detection/tracking and states estimation, such as GNSS+IMU, Lidar+IMU, and so on. Thus, some related work should be included in the introduction: yolov5-tassel: detecting tassels in rgb uav imagery with improved yolov5 based on transfer learning, an automated driving systems data acquisition and analytics platform, improved vehicle localization using on-board sensors and vehicle lateral velocity.
3) Please explain in detail how to obtain the measurement and process noise vectors for the 3D Kalman filter with valuable reference included: imu-based automated vehicle body sideslip angle and attitude estimation aided by gnss using parallel adaptive kalman filters, estimation on imu yaw misalignment by fusing information of automotive onboard sensors.
4) The curves in the figure need to be represented by different line types.
5) Work limitations and future work should be included in the conclusion.
Author Response
Thank you for your review. Below are my responses to your feedback:
1) Kalman filters are a popular method to estimate autonomous vehicle states as they are easy to follow and robust. Currently, there are a lot of filter variants to optimize the state estimation. And the host vehicle velocity is pretty important for target status update. Thus, some related work should be included: autonomous vehicle kinematics and dynamics synthesis for sideslip angle estimation based on consensus kalman filter, automated vehicle sideslip angle estimation considering signal measurement characteristic.
There are various methods that can be employed to estimate the velocity of the ego vehicle using millimeter-wave radar. Two approaches are commonly used:
- a) The velocity and angular velocity of the ego vehicle's centroid are determined by leveraging high-precision positioning information. Subsequently, the vehicle velocity at the 4D radar is computed using the external parameter matrix of the 4D radar relative to the vehicle centroid.
- b) Another method involves estimating the ego vehicle velocity at the 4D radar by analyzing the information extracted from the 4D radar point cloud.
It should be noted that since the focus of this paper is not on ego vehicle velocity estimation, a relatively simpler method is employed for traditional ego vehicle velocity estimation.
2) Based on the sensors’ configuration of autonomous vehicles, there are some alternative methods for object detection/tracking and states estimation, such as GNSS+IMU, Lidar+IMU, and so on. Thus, some related work should be included in the introduction: yolov5-tassel: detecting tassels in rgb uav imagery with improved yolov5 based on transfer learning, an automated driving systems data acquisition and analytics platform, improved vehicle localization using on-board sensors and vehicle lateral velocity.
In previous papers, relevant works in this field have been discussed, including the paper titled "3D Object Detection for Multi-frame 4D Automotive Millimeter-wave Radar Point Cloud" available at https://ieeexplore.ieee.org/document/9944629.
3) Please explain in detail how to obtain the measurement and process noise vectors for the 3D Kalman filter with valuable reference included: imu-based automated vehicle body sideslip angle and attitude estimation aided by gnss using parallel adaptive kalman filters, estimation on imu yaw misalignment by fusing information of automotive onboard sensors.
To estimate the IMU yaw misalignment, the study introduces an attitude and velocity integration method in the vehicle level frame. Additionally, the study derives the attitude error dynamics, which include yaw misalignment, pitch, and roll, and the velocity error dynamics, which encompass longitudinal and lateral velocities. By analyzing the error dynamics and observation equations, the observability of the yaw misalignment is assessed using the piece-wise constant system (PWCS) and singular value decomposition (SVD) theory.
4) The curves in the figure need to be represented by different line types.
Some changes have been made.
5) Work limitations and future work should be included in the conclusion.
Some changes have been made.
Reviewer 2 Report
A target tracking algorithm based on 4D millimeter wave radar point cloud information for autonomous driving applications is proposed in this paper. The limitations of traditional 2+1D radar systems were addressed by using higher resolution target point cloud information that enables more accurate motion state estimation and target contour information. The content in this paper is a good piece of work that is valuable for the engineering application and has the good potential to be commercialized. Thus, the publication of this work will be of interest to the readers from autonomous driving community. I have the comments below that might be helpful to improve the paper.
- Please include a figure that shows the holistic framework of the algorithm based on the 4D radar. Figure 2 and Figure 3 are subfigures of the framework. The paper lacks a big picture.
-The green boxes in Figure 1 are too large and not necessary and please improve it.
- Authors mentioned that the limitations of traditional 2+1D radar systems were addressed by using higher resolution target point cloud information. How dense is the point cloud from the 4D radar? Can you compare it with the LiDAR sensor? I saw in the experimental setup, there is a LiDAR.
- At the end of the discussion section, I suggest adding the potential application field by discussing them briefly in one or two paragraphs. This will help readers to understand the meaning of the work. Specifically, please include the works here: dynamic drifting control for general path tracking of autonomous vehicles; a hierarchical energy efficiency optimization control strategy for distributed drive electric vehicles. Because the ADAS application or vehicle dynamics control is a good application area for the radar.
Author Response
Thank you for your review. Below are my responses to your feedback:
- Please include a figure that shows the holistic framework of the algorithm based on the 4D radar. Figure 2 and Figure 3 are subfigures of the framework. The paper lacks a big picture.
Thank you for your feedback. I appreciate your suggestions. I apologize for any confusion caused by unclear descriptions in the paper. I have made revisions to address this issue. Specifically, Figure 2 provides an overview of the paper's framework, while Figure 3 is a subfigure of Figure 2. Furthermore, I have included additional explanations for some of the Figures in the article.
-The green boxes in Figure 1 are too large and not necessary and please improve it.
In order to showcase the distribution changes of the point cloud under the 3D bounding box of the target at different time points, Figure 1 was included. The green boxes represent the bounding boxes of the targets. The purpose of including this figure is to demonstrate the correlation between the point cloud distribution and the target bounding box. I have optimized Figure 1 to enhance its visual appeal.
- Authors mentioned that the limitations of traditional 2+1D radar systems were addressed by using higher resolution target point cloud information. How dense is the point cloud from the 4D radar? Can you compare it with the LiDAR sensor? I saw in the experimental setup, there is a LiDAR.
We utilized our self-collected dataset called TJ4DRadSet, which comprises point clouds from various 4D radars and LiDARs. A comparison of point cloud quantities can be observed in Figure 4 of the TJ4DRadSet paper titled "TJ4DRadSet: A 4D Radar Dataset for Autonomous Driving," available at https://arxiv.org/abs/2204.13483. In a single frame, the number of point clouds from 4D radars ranges from approximately 1000 to 6000, while LiDARs exhibit a range of 70,000 to 100,000 point clouds.
- At the end of the discussion section, I suggest adding the potential application field by discussing them briefly in one or two paragraphs. This will help readers to understand the meaning of the work. Specifically, please include the works here: dynamic drifting control for general path tracking of autonomous vehicles; a hierarchical energy efficiency optimization control strategy for distributed drive electric vehicles. Because the ADAS application or vehicle dynamics control is a good application area for the radar.
Some potential applications have been added to the discussion section of the paper.
Reviewer 3 Report
This contribution presents a method to track a moving and stationary targets based on data from a high-resolution 4D millimiter wave radar. A new approach to use the Doppler information to support the bounding box orientation estimation is introduced.
Minor problems:
Line 34: (x, y, z) should be (x, y, v)?
Line 127: Fig. number missing.
Caption of Figure 1: t-2: t should be italicized because it is a variable (please be consistent with this across the document) and "-" should be "−". - is a dash, − is a minus.
Line 252 and line 306: k should be italicized, because it is a variable.
Line 290 and line 302 repeat the same statement.
Line 348: "8." should not be bold.
Line 380: Equation (32) doesn't show how to calculate the velocity.
Line 385: This is not the system, but the measurement / observation noise.
Line 448: Not in Figure 8-9, but 11-12?
There are minor problems with the numbering of the figures / equations and with the formatting of variables. Please review the paper carefully.
Author Response
Thank you for your feedback. I have carefully reviewed the paper and made the necessary changes to address the issues with figure and equation numbering, as well as variable formatting. I appreciate your thorough review and valuable input. Thank you for reviewing my paper.